# HuRef: HUman-REadable Fingerprint for Large Language Models

## ABSTRACT

Protecting the copyright of large language models (LLMs) has become crucial due to their resource-intensive training and accompanying carefully designed licenses. However, identifying the original base model of an LLM is challenging due to potential parameter alterations through fine-tuning or continued pretraining. In this study, we introduce HuRef, a human-readable fingerprint for LLMs that uniquely identifies the base model without exposing model parameters or interfering with training. We first observe that the vector direction of LLM parameters remains stable after the model has converged during pretraining, showing negligible perturbations through subsequent training steps, including continued pretraining, supervised fine-tuning (SFT), and RLHF, which makes it a sufficient condition to identify the base model. The necessity is validated by continuing to train an LLM with an extra term to drive away the model parameters' direction and the model becomes damaged. However, this direction is vulnerable to simple attacks like dimension permutation or matrix rotation, which significantly change it without affecting performance. To address this, leveraging the Transformer structure, we systematically analyze potential attacks and define three invariant terms that identify an LLM's base model. We make these invariant terms human-readable by mapping them to a Gaussian vector using a convolutional encoder and then converting it into a natural image with StyleGAN2. The encoder discriminates between invariants from different base models and ensures Gaussian output through adversarial training, while StyleGAN2 transforms Gaussian vectors into dog images. Consequently, our method generates a dog image as an identity fingerprint for an LLM, where the dog's appearance strongly indicates the LLM's base model. Specifically, if the LLM is adapted from another base model, the generated dog highly resembles that model; otherwise if trained independently from scratch, it exhibits a unique dog image distinct from other models. Experimental results across various LLMs demonstrate the effectiveness of our method, the generated dog image remains invariant to different training steps, including SFT, RLHF, or even continued pretraining with augmented vocabulary in a new language.

## 1 INTRODUCTION

Large language models (LLMs) have become the foundation models in many scenarios of artificial intelligence. As training an LLM from scratch consumes a huge amount of computation and data resources and the trained LLM needs to be carefully protected from malicious use, the parameters of the LLMs become a crucial property to protect, for both commercial and ethical reasons. As a result, many of the LLMs are open-sourced with carefully designed licences to reject commercial use(Touvron et al., 2023a)(Taylor et al., 2022) or requiring an apply-and-approval process(Touvron et al., 2023b)(Zhang et al., 2022)(Penedo et al., 2023)(BaiChuan-Inc, 2023)(Team, 2023)(Zheng et al., 2023b), let alone some LLMs are not open-sourced entirely(OpenAI, 2022)(GPT-4, 2023)(Brown et al., 2020)(Wu et al., 2023b)(Chowdhery et al., 2022)(Hoffmann et al., 2022).

At the core of protecting LLMs from unauthorized use is to identify the base model of a given LLM. However, different from other forms of property such as software or images, protecting LLMs is a novel problem with unique challenges. First, the base model usually needs to be fine-tuned or even continue pretrained to be applied to downstream tasks, resulting in parameter updates that make

the resulting model different from the original base model, which makes it disputable to identify the base model. Second, many of the popular LLMs are not releasing their parameters, leaving the identification in a black-box setting. Third, different from previous smaller-scale neural networks that are only trained for specific tasks, LLMs are usually targeted for enormous forms of tasks that are not yet defined during pretraining. This has made the watermarking methods for traditional neural networks(Adi et al., 2018)(Xiang et al., 2021)(Yadollahi et al., 2021) not suited in this case, especially under extensive subsequent training steps.

In this work, we propose a novel way to overcome the aforementioned challenges by proposing a model that reads part of the model parameters and computes a fingerprint for each LLM, without directly exposing the LLM parameters to the public or interfering with its training process. The appearance of the fingerprint is closely dependent on the base model, and invariant to almost all subsequent training steps, including supervised fine-tuning (SFT), reinforcement learning with human feedback (RLHF), or even continue-pretraining with augmented vocabulary in a new language.

The fingerprint is based on our observation that the vector direction of LLM parameters remains stable against various subsequent training steps after the model has converged during pretraining. This makes it a good indicator for base model identification. Empirically, the sufficiency of this correlation is elaborated in Section 3.1.1, while its necessity is presented in Section 3.1.2.

Further, despite its stability towards training, the vector direction of the model parameter is vulnerable to some simple direct weight rearrangements that could significantly change the direction of parameter vectors without affecting the model's performance. We construct three invariant terms that are robust to these weight rearrangements by systematically analyzing possible rearrangements and leveraging the Transformer structure. This is elaborated in Section 3.2.

Moreover, we make the fingerprint human-readable by mapping the invariant terms into a Gaussian random vector through a convolutional encoder and then map the Gaussian vector to a natural image through an off-the-shelf image generation model, StyleGAN2(Karras et al., 2020). This makes our fingerprints easily interpretable and straightforward to decipher. This is elaborated in Section 4.

With this fingerprinting approach, we can sketch an outline for protecting LLMs (c.f. Appendix B).

## 2 RELATED WORKS

Despite its short history, safeguarding LLMs against unauthorized use has been a topic of significant interest. There are two primary categories of approaches.

**Post-hoc Detection** methods involve analyzing text generated by LLMs after its production. LLMDet (Wu et al., 2023a) calculates proxy perplexity by leveraging prior knowledge of the model's next-token probabilities. DetectGPT (Mitchell et al., 2023) uses model-predicted probabilities to identify passages generated by a specific LLM. Li et al. (2023) employs perplexity scores and intricate feature engineering. These methods are usually applicable to a specific LLM and could be affected by supervised fine-tuning (SFT) and continued pretraining. More recently, Sadasivan et al. (2023) presented theoretical findings that for highly advanced AI human mimickers, even the best possible detection methods may only marginally outperform random guessing.

**Watermarking Techniques** can be divided into two main categories (Boenisch, 2021). The first embeds watermarks or related information into the model parameters, such as explicit watermarking scheme (Uchida et al., 2017) or leveraging another neural network (Wang et al., 2020), which could potentially affect model performance (Wang & Kerschbaum, 2019). The second category focuses on inducing unusual prediction behavior in the model. Xiang et al. (2021) explored embedding phrase triggers, and Gu et al. (2022) extended this approach to LLMs, albeit they are task-specific. Yadollahi et al. (2021) proposed a watermarking method but did not consider subsequent fine-tuning. Christ et al. (2023) proposed cryptographic designs. Kirchenbauer et al. (2023) involved using preselected tokens, but this inevitably alters the model prediction. These methods may turn out to be vulnerable to attacks on certain tokens, for example, Krishna et al. (2023) successfully evaded watermarking (Kirchenbauer et al., 2023), GPTZero(Tian, 2023), DetectGPT, and OpenAI's text classifier(OpenAI, 2023) using paraphrasing attacks.

Our work doesn't fall into any of the two categories since it is based on analyzing model weights post-hoc and relies on a wide spectrum of tokens.

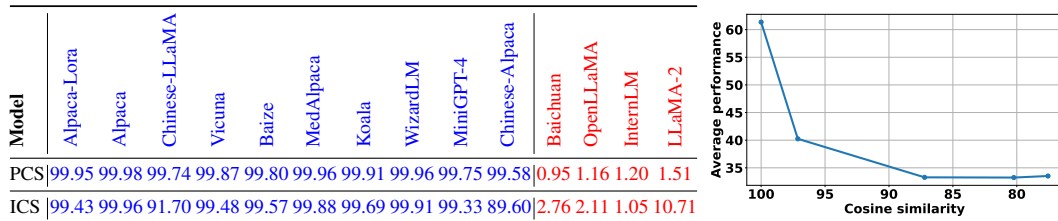

| Model | Alpaca-Lora | Alpaca | Chinese-LLaMA | Vicuna | Baize | MedAlpaca | Koala | WizardLM | MiniGPT-4 | Chinese-Alpaca | Baichuan | OpenLLaMA | InternLM | LLaMA-2 |
|---|---|---|---|---|---|---|---|---|---|---|---|---|---|---|
| PCS | 99.95 | 99.98 | 99.74 | 99.87 | 99.80 | 99.96 | 99.91 | 99.96 | 99.75 | 99.58 | 0.95 | 1.16 | 1.20 | 1.51 |
| ICS | 99.43 | 99.96 | 91.70 | 99.48 | 99.57 | 99.88 | 99.69 | 99.91 | 99.33 | 89.60 | 2.76 | 2.11 | 1.05 | 10.71 |

Table 1: The cosine similarities of model parameters (PCS) and invariant terms (ICS) between various LLMs w.r.t. the LLaMA-7B base model. All models are of the same size.

Figure 1: The model's performance quickly deteriorates as the PCS decreases.

## 3 VECTOR DIRECTION OF LLM PARAMETERS AND THE INVARIANT TERMS

### 3.1 USING VECTOR DIRECTION OF LLM PARAMETERS TO IDENTIFY THE BASE MODEL

We can flatten all weight matrices and biases of an LLM into vectors, concatenate all resulting vectors together, and treat is as single huge vector. In this subsection, we are going to show how the direction of this vector could be used to determine the base model by empirically showing its sufficiency and necessity.

#### 3.1.1 SUFFICIENCY

For sufficiency, we compute the cosine similarities between a base model LLaMA-7B and various of its offspring models, as well as other independently pretrained LLMs(Geng & Liu, 2023) that are of the same size. Table 1 shows a wide spectrum of models that inherit the LLaMA-7B base model, whose subsequent training processes involve various training paradigms, such as SFT (Taori et al., 2023)(Xu et al., 2023b)(Zheng et al., 2023a)(Geng, 2023)(Xu et al., 2023a)(Han et al., 2023), SFT with LoRA (Wang, 2023) and extensive continue pretraining in a new language(Cui et al., 2023), extending to new modalities (Zhu et al., 2023), etc. Please refer to Appendix C for a detailed description of the subsequent training setting of these models.

Regardless of their various subsequent training setting, we can figure that all of these models show almost full scores in cosine similarity, largely preserving the base model's parameter vector direction. On the other hand, the models that are trained independently appear to be completely different in parameter vector direction, showing almost zero cosine similarity with the LLaMA-7B model.

These observations indicate that a high cosine similarity between the two models highly suggests that they share the same base model, and vice versa.

#### 3.1.2 NECESSITY

From the necessity perspective, we want to verify that if the base model's ability could still be preserved when the cosine similarity is intentionally suppressed in subsequent training steps. To this end, we inherit the LLaMA-7B base model and interfere with the Alpaca's SFT process by augmenting the original SFT loss with an extra term that minimizes the absolute value of cosine similarity. i.e. $L_A = \frac{|\langle \boldsymbol{V}_A, \boldsymbol{V}_{base} \rangle|}{|\boldsymbol{V}_A||\boldsymbol{V}_{base}|}$. Here $\boldsymbol{V}_A, \boldsymbol{V}_{base}$ stand for the parameter vector of the model being tuned and that of the base model, respectively.

Figure 1 presents the average zero-shot performance on a set of standard benchmarks when $L_A$ is at different values. The benchmarks include BoolQ(Clark et al., 2019), PIQA(Bisk et al., 2020), HellaSwag(Zellers et al., 2019), WinoGrande(Sakaguchi et al., 2021), ARC-e, ARC-c(Clark et al., 2018),RACE(Lai et al., 2017) and MMLU(Hendrycks et al., 2020). (c.f. Appendix E for a detailed breakdown of performances on each task.) We can see that despite the original training loss is still present, the model quickly deteriorates to random guesses as the cosine similarity detaches away from that of the base model.

These observations indicate that it is fairly hard for the model to preserve the base model's performance without keeping a high cosine similarity to it.

## 3.2 Deriving the Invariant Terms

Although the vector direction of model parameters is shown to closely stick to its base model, directly comparing the vector direction through cosine similarity requires both models to reveal their parameters, which is unacceptable in many cases. In addition, apart from training, parameter vector direction is vulnerable to some attacks that directly rearrange the model weights. For example, since the hidden units in a model layer are permutation-invariant, one can easily alter the parameter vector direction by randomly permuting the hidden units along with the weights wired to the units. These attacks are invisible to discover since they could easily break the cosine similarity but neither change the model structure nor affect the model performance.

In this subsection, we are going to first systematically analyze and formalize possible weight rearrangements by leveraging the structure constraints of Transformer, and then derive three terms that are invariant under these rearrangements, even when they are combined.

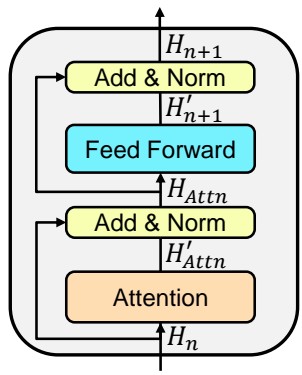

Figure 2: Transformer layer

Let's first consider the Transformer layer as depicted in Figure 2. Formally, the layer conducts the following computation:

$$H'_{Attn} = \text{softmax}\left(\frac{H_n W_Q (H_n W_K)^T}{\sqrt{d}}\right) H_n W_V W_O \quad (1)$$

$$H'_{n+1} = \sigma\left(H_{Attn} W_1 + \mathbf{b}_1\right) W_2 + \mathbf{b}_2 \quad (2)$$

where $H_n \in \mathbb{R}^{l \times d}$ is the hidden state of the $n$-th layer, with $l, d$ being sequence length and model dimensions, respectively. $H'_{Attn}$ is the self-attention output. To reduce clutter, we omit equations related to residual connection and LayerNorm, but denote the variables right before it with an apostrophe. The $W$'s and $\mathbf{b}$'s are weights and biases.

Note that the first layer reads the word embedding, i.e., $H_0 = X \in \mathbb{R}^{l \times d}$, and the final output distribution $\mathbf{P} \in \mathbb{R}^{l \times v}$ is given by

$$\mathbf{P} = \text{softmax}\left(H_N E\right) \quad (3)$$

where $v$ is the vocabulary size, $N$ is the total number of layers, and $E \in \mathbb{R}^{d \times v}$ is the parameter matrix in the softmax layer, which is sometimes tied with the word embedding matrix at the input.

### 3.2.1 Forms of Weight Rearrangement Attacks

Putting Equations 1~3 together, we can systematically analyze how the parameter vector direction can be attacked through direct weight rearrangements. There are totally 3 forms of attacks that could camouflage the model without changing its architecture or affecting its output.

**1. Linear mapping attack on $W_Q, W_K$ and $W_V, W_O$.** Consider Equation 1, one can transform $W_Q$ and $W_K$ symmetrically so that the product $W_Q W_K^T$ remains unchanged but both weights are significantly modified. This will alter the parameter vector direction significantly. Formally, for any invertible matrix $C_1$, let

$$\tilde{W}_Q = W_Q C_1, \quad \tilde{W}_K = W_K C_1^{-1} \quad (4)$$

and substitute them respectively into the model, one can camouflage it as if it's a brand new model, without sacrificing any of the base model's performance. The same holds for $W_V, W_O$ as well.

**2. Permutation attack on $W_1, \mathbf{b}_1, W_2$.** Consider Equation 2, since it consists of two fully connected layers, one can randomly permute the hidden states in the middle layer without changing its output. Formally, let $P_{FFN}$ be an arbitrary permutation matrix, one can camouflage the model without sacrificing its performance by substituting the following three matrices accordingly

$$\tilde{W}_1 = W_1 P_{FFN}, \quad \tilde{W}_2 = P_{FFN}^{-1} W_2, \quad \tilde{\mathbf{b}}_1 = \mathbf{b}_1 P_{FFN} \quad (5)$$

**3. Permutation attack on word embeddings.** In a similar spirit, one can permute the dimensions in the word embedding matrix as well, although it would require all remaining parameters to be permuted accordingly. Formally, let $P_E$ be an arbitrary permutation matrix that permutes the dimensions in $X$ through $\tilde{X} = X P_E$, due to the existence of the residual connections, the output of

all layers have to be permuted in the same way, i.e., $\tilde{H}_n = H_n P_E$. Note that it's not necessarily the case in the former two types of attacks. This permutation has to be canceled out at the final softmax layer (Equation 3), by permuting the dimensions in $E$ accordingly, i.e. $\tilde{E} = P_E^{-1} E$. Specifically, all remaining parameters have to be permuted in the following way:

$$\tilde{W}_Q = P_E^{-1} W_Q, \quad \tilde{W}_K = P_E^{-1} W_K, \quad \tilde{W}_V = P_E^{-1} W_V, \quad \tilde{W}_O = W_O P_E$$
$$\tilde{W}_1 = P_E^{-1} W_1, \quad \tilde{W}_2 = W_2 P_E, \quad \tilde{\mathbf{b}}_2 = \mathbf{b}_2 P_E \tag{6}$$

Moreover, putting everything together, one can combine all the aforementioned three types of attacks altogether. Formally, the parameters can be camouflaged as:

$$\tilde{W}_Q = P_E^{-1} W_Q C_1, \quad \tilde{W}_K = P_E^{-1} W_K C_1^{-T}, \quad \tilde{W}_V = P_E^{-1} W_V C_2, \quad \tilde{W}_O = C_2^{-1} W_O P_E$$
$$\tilde{W}_1 = P_E^{-1} W_1 P_{FFN}, \quad \tilde{\mathbf{b}}_1 = \mathbf{b}_1 P_{FFN}, \quad \tilde{W}_2 = P_{FFN}^{-1} W_2 P_E, \quad \tilde{\mathbf{b}}_2 = \mathbf{b}_2 P_E$$
$$\tilde{X} = X P_E, \quad \tilde{E} = P_E^{-1} E$$
$$\tag{7}$$

Note that for permutation matrix we have $P^{-1} = P^T$. This includes all possible attacks that 1) do not change the model architecture, and 2) do not affect the model's output.

### 3.2.2 THE INVARIANT TERMS TO THESE ATTACKS

In order to find the invariant terms under all these attacks, we need to combine terms in Equation 7 to get the invariant term that nicely cancels out all extra camouflaging matrices. To this end, we construct 3 invariant terms:

$$M_a = \hat{X} W_Q W_K^T \hat{X}^T, \quad M_b = \hat{X} W_V W_O \hat{X}^T, \quad M_f = \hat{X} W_1 W_2 \hat{X}^T \tag{8}$$

Note that we are not simply including all tokens of a vocabulary or tokens in a certain sentence for $X$, and instead, we use $\hat{X}$ in these terms. There are two problems if we directly use all tokens' embeddings $X$. First, using the whole embedding matrix will make the terms unnecessarily large and of variable size between different models. Second, more importantly, since it is fairly common to inherit a base model with an augmented vocabulary, i.e., to append a set of new tokens at the end of the original vocabulary, the invariant terms will result in different sizes and are not comparable in this case. Third, if we designate specific tokens instead, the selected tokens may not always exist in all LLMs being tested. As a result, we carefully choose the tokens to be included in $\hat{X}$, by following these steps:

1. Select a sufficiently big corpus as a standard verifying corpus.
2. Tokenize the corpus with the LLM's vocabulary, and sort all tokens in the vocabulary according to their frequency.
3. Delete all tokens in the vocabulary that don't show up in the corpus.
4. Among the remaining tokens, select the least frequent $K$ tokens as the tokens to be included in $\hat{X}$.

Here, using a standard corpus ensures that the resulting tokenization will be identical if a certain model's vocabulary is a subset of another; the sufficiently large corpus stabilizes the frequencies of tokens in the vocabulary and provides enough chance for as many tokens as possible to show up. Deleting zero-shot tokens automatically sweeps off augmented tokens. Selecting the rarest tokens minimizes potential affections brought by parameter updates in subsequent training processes. A properly large $K$ will ensure a large enough set of tokens are included, making the resulting invariant terms more generally representative. More importantly, it will make all the invariant terms have the same size across all LLMs, regardless of their original sizes. In practice, we choose $K = 4096$.

We show the cosine similarity between the invariant terms in Table 1. Although not as perfect as the parameter's cosine distances, they still preserve a high correlation to the base model.

## 4 MAKING THE INVARIANT TERMS HUMAN-READABLE: THE FINGERPRINTING MODEL

Instead of directly using the three invariant terms, we can present the content in the terms in a human-readable way, through the fingerprinting model. The fingerprinting model consists of a neu-

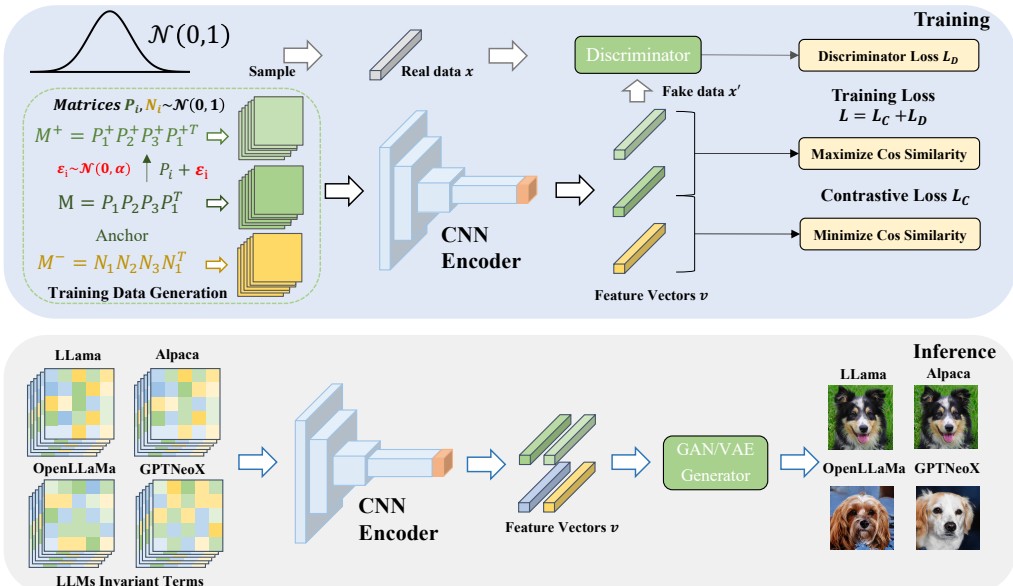

Figure 3: The training and inference of our fingerprinting model.

ral network encoder - a convolutional encoder in our case - and an off-the-shelf image generator as depicted in Figure 3. In principle, the encoder takes as input the invariant terms of a certain model, tile them together, and deterministically maps them to a vector that appears to be filled with Gaussian variables. The subsequent image generator reads this vector and maps it to a natural image. Importantly, throughout the process, the locality of the inputs has to be preserved from end to end. i.e., similar invariable terms should result in similar Gaussian variables and finally similar images.

## 4.1 TRAINING THE CONVOLUTIONAL ENCODER

All the invariant terms in Equation 8 have the same size, i.e., $M_a, M_b, M_f \in \mathbb{R}^{K \times K}$, regardless of the index of the layer or LLM sizes. As a result, we can tile up them to form a 3D input tensor $M \in \mathbb{R}^{K \times K \times C}$, where $C$ is the channel dimension. If we are using all layers, $C = 3N$. Again, in order to make $M$ the same size across all models, we only involve the last $r$ layers in the LLM[1].

Note that we don't need to use any real LLM weights for training the convolutional encoder, as it only needs to learn a locality-preserving mapping between the input tensor and the output Gaussian vector. This ensures strict exclusivity between the training and test data. To construct the training data, we synthesize the matrix in each channel of $M$ on-the-fly, by randomly sampling 3 matrices $P_1, P_2, P_3$ and multiplying them together as $P_1 P_2 P_3 P_1^T$, as though they are model parameters.

To learn locality-preserving mapping, we adopt contrastive learning. For a randomly sampled input $M$, its negative sample is given by another independently sampled tensor $M^-$. For its positive sample $M^+$, we perturb the content in each of $M$'s channel by adding small perturbation noises $\epsilon_i \in \mathcal{N}(0, \alpha)$ to the 3 matrices behind it. Here $\alpha$ is a hyperparameter determining the small variance. Subsequently, the contrastive loss $L_C$ is given by:

$$L_C = \left|(1 - S_C(M, M^+))\right| + \left|S_C(M, M^-)\right| \tag{9}$$

where $S_C(\cdot, \cdot)$ computes the cosine similarity between its two input matrices.

To render the output vector appears to be Gaussian, we adopt the standard GAN(Karras et al., 2019) training scheme. We add a simple MLP as the discriminator $D$ that is trained to discriminate between real Gaussian vectors and the convolutional output vector $v$. In this setting, the convolutional encoder serves as the generator. During training, for every $m$ steps, we alternate between training the discriminator and the generator. The discriminator loss $L_D$ is thus given by

---

[1]In fact, experimentally we find that a small $r$ is already sufficient to discriminate LLMs, it's not necessary to involve many layers. In all of our experiments, $r = 2$, so there are only 6 channels in the input.

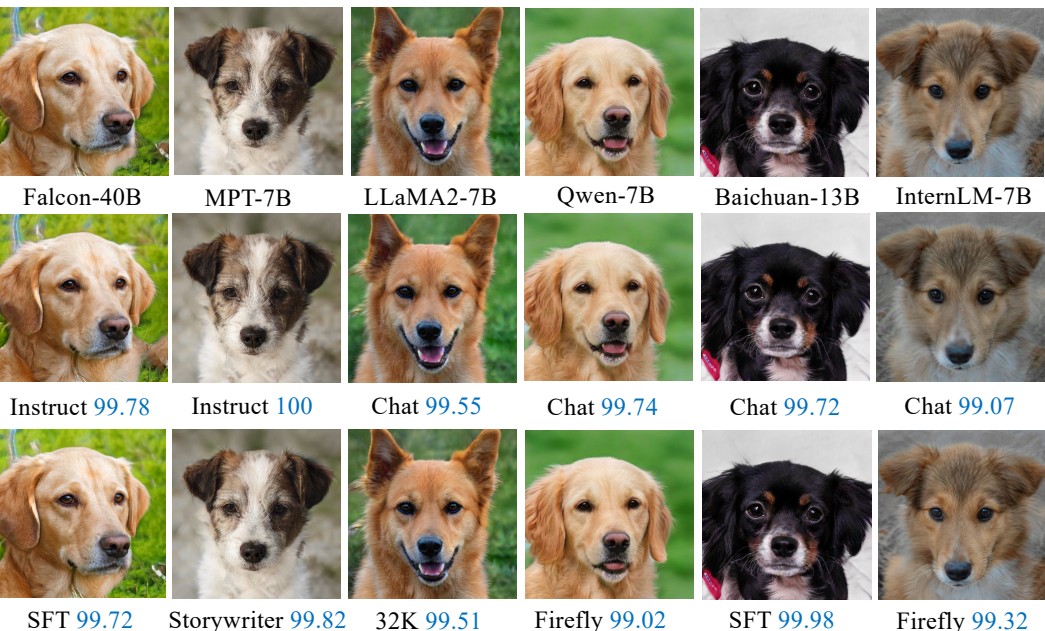

Figure 4: Fingerprints of 6 different base models (in the first row) and their corresponding offspring models (the lower two rows) are presented. The base model's name is omitted in the offspring models. The blue number to the right of each offspring model indicates the cosine similarity of its invariant terms (ICS) w.r.t. its base model.

$$L_D = \frac{1}{m} \sum_{i=1}^{m} \log\left(1 - D\left(\boldsymbol{v}\right)\right) \tag{10}$$

While training the generator we also need to incorporate the contrastive learning loss. Thus the actual loss $L$ for the training generator is a combination of $L_C$ and $L_D$.

$$L = L_C + L_D \tag{11}$$

## 4.2 INFERENCE

In the inference stage, the convolutional encoder takes the invariant terms from real LLMs and outputs $\boldsymbol{v}$. In principle, any image generator that takes a Gaussian input and has the locality-preserving property would fit here. In this paper, we employ the StyleGAN2 generator pre-trained on the AFHQ(Choi et al., 2020) dog dataset to generate natural images, we detailed why it's a smooth generator in Appendix D. By visually representing the invariant terms as fingerprints, we can easily identify base models based on their fingerprint images, enabling reliable tracking of model origins.

## 5 EXPERIMENTS

Apart from the results shown in Section 3, our experiment section is three-fold. First, we choose several widely used and open-sourced LLM base models with their offspring LLMs to compute their invariant terms and fingerprint images. Second, we extensively test our method on the LLaMA model family, including some heavily continue-pretrained ones to experimentally verify the robustness of our method against subsequent training processes. Third, we extensively experimented on more open-sourced LLMs to show the diversity of generated images on different LLMs.

### 5.1 INDEPENDENTLY TRAINED LLMS AND THEIR OFFSPRING MODELS

We conduct experiments on 6 commonly used open-sourced LLMs with their sizes ranging from 7B to 40B. For the invariant terms, we use the last 2 layers' parameters and set $K = 4096$, which results

| ICS | LLaMA | MiGPT | Alpaca | MAlpaca | Vicuna | Wizard | Baize | AlpacaL | CAlpaca | Koala | CLLaMA | Beaver | Guanaco | BiLLa |
|-----|-------|-------|--------|---------|--------|--------|-------|---------|---------|-------|--------|--------|---------|-------|
| LLaMA | 100.00 | 99.33 | 99.96 | 99.88 | 99.48 | 99.91 | 99.57 | 99.43 | 89.60 | 99.69 | 91.70 | 99.98 | 91.68 | 81.13 |
| MiGPT | 99.33 | 100.00 | 99.30 | 99.23 | 99.20 | 99.27 | 98.93 | 98.76 | 89.02 | 99.14 | 91.10 | 99.32 | 91.15 | 80.76 |
| Alpaca | 99.96 | 99.30 | 100.00 | 99.84 | 99.45 | 99.88 | 99.53 | 99.40 | 89.57 | 99.66 | 91.67 | 99.98 | 91.66 | 81.09 |
| MAlpaca | 99.88 | 99.23 | 99.84 | 100.00 | 99.38 | 99.80 | 99.46 | 99.31 | 89.49 | 99.58 | 91.60 | 99.86 | 91.57 | 81.03 |
| Vicuna | 99.48 | 99.20 | 99.45 | 99.38 | 100.00 | 99.41 | 99.08 | 98.92 | 89.16 | 99.25 | 91.25 | 99.47 | 91.30 | 80.93 |
| Wizard | 99.91 | 99.27 | 99.88 | 99.80 | 99.41 | 100.00 | 99.49 | 99.35 | 89.52 | 99.63 | 91.62 | 99.89 | 91.60 | 81.09 |
| Baize | 99.57 | 98.93 | 99.53 | 99.46 | 99.08 | 99.49 | 100.00 | 99.05 | 89.21 | 99.27 | 91.30 | 99.55 | 91.32 | 80.79 |
| AlpacaL | 99.43 | 98.76 | 99.40 | 99.31 | 98.92 | 99.35 | 99.05 | 100.00 | 89.08 | 99.12 | 91.17 | 99.42 | 91.27 | 80.70 |
| CAlpaca | 89.60 | 89.02 | 89.57 | 89.49 | 89.16 | 89.52 | 89.21 | 89.08 | 100.00 | 89.34 | 97.58 | 89.58 | 82.81 | 73.11 |
| Koala | 99.69 | 99.14 | 99.66 | 99.58 | 99.25 | 99.63 | 99.27 | 99.12 | 89.34 | 100.00 | 91.43 | 99.67 | 91.37 | 80.94 |
| CLLaMA | 91.70 | 91.10 | 91.67 | 91.60 | 91.25 | 91.62 | 91.30 | 91.17 | 97.58 | 91.43 | 100.00 | 91.68 | 84.33 | 74.73 |
| Beaver | 99.98 | 99.32 | 99.98 | 99.86 | 99.47 | 99.89 | 99.55 | 99.42 | 89.58 | 99.67 | 91.68 | 100.00 | 91.67 | 81.11 |
| Guanaco | 91.68 | 91.15 | 91.66 | 91.57 | 91.30 | 91.60 | 91.32 | 91.27 | 82.81 | 91.37 | 84.33 | 91.67 | 100.00 | 75.62 |
| BiLLa | 81.13 | 80.76 | 81.09 | 81.03 | 80.93 | 81.09 | 80.79 | 80.70 | 73.11 | 80.94 | 74.73 | 81.11 | 75.62 | 100.00 |

Table 2: The cosine similarities of invariant terms between various pairs of LLaMA-7B and its off-spring models. Abbreviations used include "MAlpaca" for "MedAlpaca," "AlpacaL" for "Alpaca-Lora," "MiGPT" for "MiniGPT-4," "Wizard" for "WizardLM," and "CAlpaca" and "CLLaMA" representing "Chinese-Alpaca" and "Chinese-LLaMA," respectively.

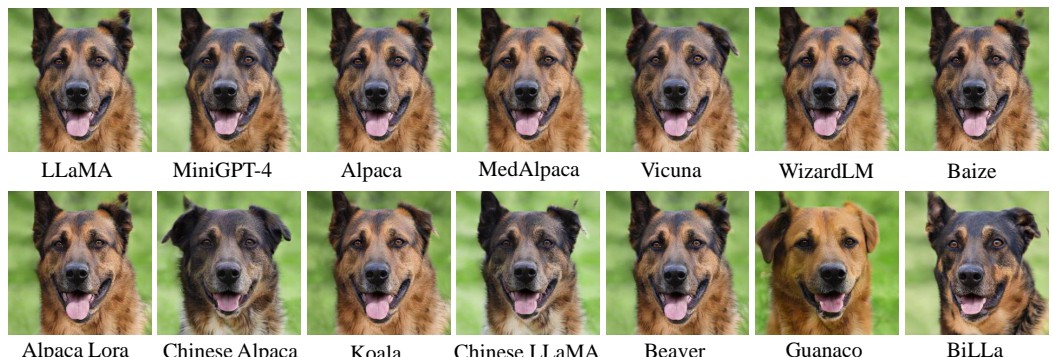

Figure 5: Fingerprints of LLaMA-7B and its offspring models.

in $M \in \mathbb{R}^{4096 \times 4096 \times 6}$ for all models. The 6 base models are Falcon-40B, MPT-7B(Lin et al., 2022), LLaMA2-7B, Qwen-7B (Bai et al., 2023), Internlm-7B, and Baichuan-13B. Most of their offspring models went through an SFT process, except for LLaMA2-7B-32K(Together.ai, 2023), which also had a continued pretraining before SFT. Please refer to Appendix F for a detailed description of the subsequent training steps each model has undergone. Apart from the fingerprints (Figure 4) we also calculate, for each offspring model, its invariant terms' cosine similarity (ICS) w.r.t. its base model. Remarkably, for all the offspring models, their fingerprints closely resemble those of their base models, and the ICS indicates the high similarity of their invariant terms w.r.t. their base models. On the other hand, LLMs based on different base models yield fingerprints that are highly different, covering diverse appearances and breeds of dogs.

## 5.2 LLaMA and its offspring models

To extensively validate how our proposed method is robust to various subsequent training processes, we choose the LLaMA-7B base model as a testbed, since it is a widely used model with the largest family of offspring models. We collect 10 offspring models introduced in Section 3.1.1. Plus Beaver which underwent RLHF(Dai et al., 2023), Guanaco(Dettmers et al., 2023) SFT on the multilingual dataset, and BiLLa(Li, 2023) continued pretraining on a new language. They are subject to various training paradigms, and we refer the readers to Appendix C for detailed descriptions. Among these offspring models, We want to draw attention to the Chinese-LLaMA and Chinese-Alpaca models, both of which have an extended vocabulary and have been continued pretrained for 20GB of text in a new language, with the latter trained with several millions of additional SFT samples. Following the settings in Section 5.1, the invariant terms again use the last 2 layers' parameters.

Despite all these complex subsequent training steps, all the models yield highly relevant invariant terms (Table 2). We extensively compute the cosine similarity of the invariant terms (ICS) between

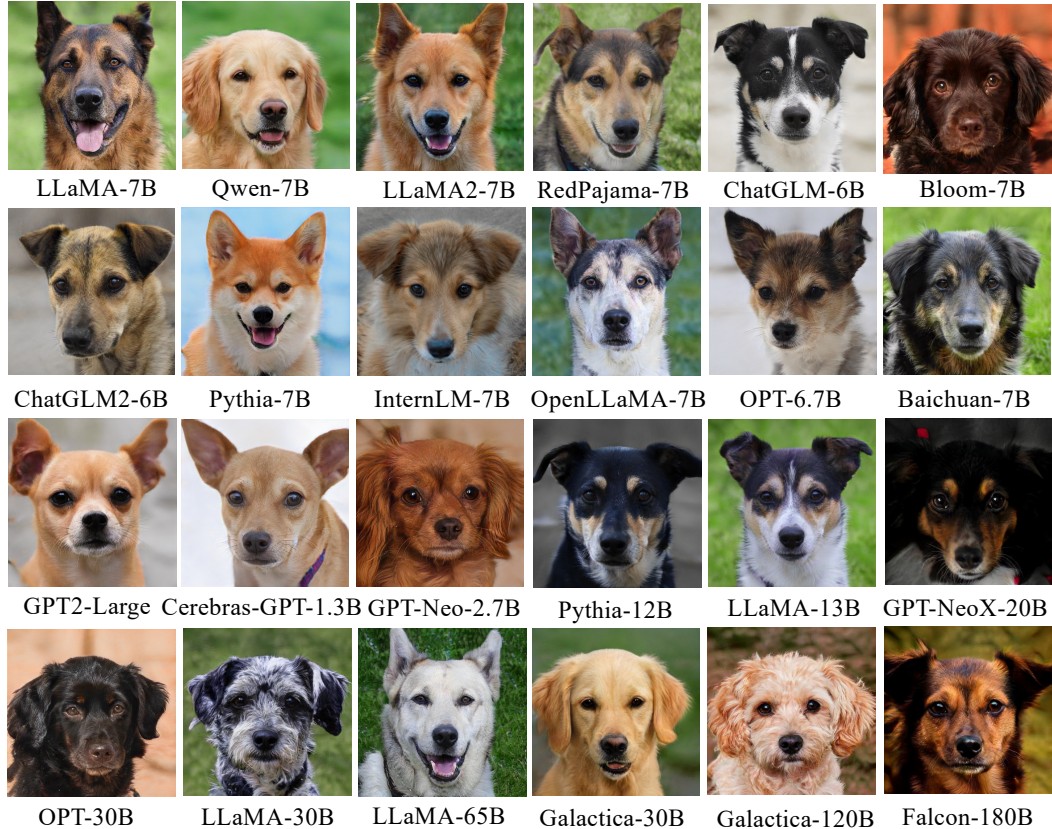

Figure 6: Fingerprints of 24 independently trained LLMs.

every pair of models and found that they are still highly similar, with the lowest ICS being 73.11. For our trained fingerprinting model, they are aligned to a similar fingerprint image of a German Shepherd, with similar poses, coat patterns, expressions, and backgrounds (Figure 5).

### 5.3 A WIDER COLLECTION OF INDEPENDENTLY TRAINED LLMs

We collect a total of 24 open-sourced LLMs with their sizes ranging from 774M (GPT2-Large) to 180B (Falcon-180B). Please refer to Appendix H for a detailed list and description of these models. We generate their fingerprints, as shown in Figure 6. The similarities between different models were notably low, further confirming the validity of our proposed method and model. Due to space limit, the ICSs between the 24 models are shown in Appendix I. We find that the ICS might accidentally become high between models with different sizes. However, as it is not possible to inherit a model's parameters into a different-sized model, these accidentally high ICSs don't affect the effectiveness of our method.

## 6 CONCLUSION

In this paper, we introduce a novel approach that uses model parameters to yield a fingerprint for LLMs. This approach could provide a human-readable identity for LLM without exposing its parameters or interfering with its training process. Different from the well-studied black-box or white-box settings, our method lies in the middle of both. Although generating the invariant terms needs access to the model parameters, which has to be done by the LLM owners and might need regulations to prevent fake terms, all the subsequent steps could be done by the public, without requiring a special third-party institution. Moreover, as only the invariant terms and fingerprint images are published, no model weights need to be exposed outside of its owner, neither to the public nor the other LLM owners, during the whole process.

## 7 ETHICS STATEMENT

This study primarily focuses on LLM protection, an area crucial for safeguarding intellectual property and preventing misuse. The collection and use of data in our research comply with all relevant privacy and data protection regulations. We do not engage in any data practices that compromise the privacy or rights of individuals.

## 8 REPRODUCIBILITY STATEMENT

We have provided detailed methodologies for our approach throughout this paper. For enhanced clarity, we have included an illustrative framework for LLM protection with fingerprints in Appendix B to help readers better grasp our intuition. Our methods are thoroughly described in Section 3, and we provide a comprehensive account of our model in Section 4. In addition, we have included information about the various LLMs we used in Appendix. These details will make it more feasible for other researchers to replicate our work and validate our findings.

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

# APPENDICES

## A IMPLEMENTATION DETAILS

### A.1 DATA SYNTHESIS

As illustrated in Figure 3, we dynamically obtain anchor, positive, and negative data on the fly by sampling matrices from a normal distribution and multiplying them following the format of the invariant terms. Following is the detailed process.

For the anchor data $M$: Sample matrices $P_1, P_2, P_3$ from a standard normal distribution. Consider $P_1$ as $\hat{X}$, and $P_2, P_3$ as model parameter matrices,

$$M = P_1 P_2 P_3 P_1^T \tag{12}$$

For positive data $M^+$: Independently sample noises $\epsilon_i$ from a normal distribution $\mathcal{N}(0, \alpha)$,

$$P_i^+ = P_i + \epsilon_i, \quad M^+ = P_1^+ P_2^+ P_3^+ P_1^{+T} \tag{13}$$

For negative data $M^-$: Independently sample matrices $N_1, N_2, N_3$ from a standard normal distribution,

$$M^- = N_1 N_2 N_3 N_1^T \tag{14}$$

### A.2 TRAINING SETTINGS

In the training stage, we alternate training the discriminator and CNN encoder every 10 steps. We set the batch size to 10, the initial learning rate to 0.0001, and introduce a noise intensity $\alpha$ of 0.16 for positive samples. After 8 epochs of training, we obtained the CNN encoder used in our paper.

### A.3 MODEL ARCHITECTURE

For the CNN encoder: The CNN encoder takes invariant terms $M \in \mathbb{R}^{4096 \times 4096 \times 6}$ as input and produces a feature vector $v$ as output. Our CNN encoder structure, as depicted in Figure 3, consists of the first four convolutional layers and the last mean pooling layer. The mean pooling layer simply calculates the average of the feature maps obtained from each channel, resulting in a feature vector

| CNN Layers | Input Channel | Output Channel | Kernel Size | Stride | Padding |
|:---:|:---:|:---:|:---:|:---:|:---:|
| Layer 1 | 6 | 8 | 48 | 4 | 22 |
| Layer 2 | 8 | 64 | 48 | 4 | 22 |
| Layer 3 | 64 | 256 | 48 | 4 | 22 |
| Layer 4 | 256 | 512 | 48 | 4 | 22 |

Table 3: Detailed hyperparameters of the stacked four convolutional layers.

$v$ with a length equal to the number of channels. The hyperparameters for the four convolutional layers are provided in the table below:

For the discriminator: We utilize a simple 3-layer MLP as the discriminator. The 512-dimensional feature vector $v$ from the CNN encoder serves as fake data, while a 512-dimensional vector $x$ sampled from the standard normal distribution serves as real data. The discriminator processes $v$ and $x$, progressively reducing dimensionality through three linear layers, and finally outputs the probability of a sample being real after applying a sigmoid activation function. The sizes of the three linear layers are $W_1 \in \mathbb{R}^{512 \times 256}$, $W_2 \in \mathbb{R}^{256 \times 128}$, and $W_3 \in \mathbb{R}^{128 \times 1}$, respectively.

For the image generator: The pre-trained StyleGAN2 checkpoint we used can be found at:

```
https://nvlabs-fi-cdn.nvidia.com/stylegan2-ada-pytorch/
pretrained/afhqdog.pkl
```

## B  AN ILLUSTRATIVE FRAMEWORK FOR LLM PROTECTION WITH FINGERPRINTS

An example framework is depicted in Figure 7.

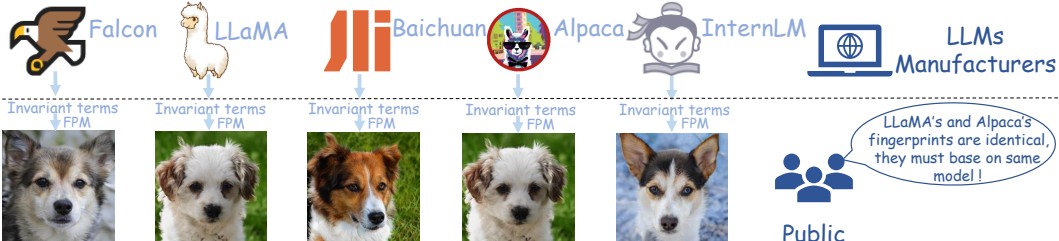

Figure 7: An illustrative framework for LLM protection with fingerprints. The LLM manufacturers compute the invariant terms internally, feed them to the public fingerprinting model (FPM) to generate a fingerprint image, and release both to the public. This allows the public to detect shared base models merely from the fingerprint image. Although manufacturers can attempt adjustments on their LLM adversarial to the public FPM to make the fingerprint appear differently, the mere use of random matrices in training the FPM enables the public to train new FPMs whenever necessary, preventing such interference. Importantly, manufacturers only disclose the invariant terms and fingerprint images, without revealing model parameters or affecting LLM training.

## C  DETAILED DESCRIPTION OF THE LLAMA FAMILY OF MODELS

In Table 1, we calculate the cosine similarities of model parameters between various LLMs w.r.t. the LLaMA-7B base model. Among these models, Alpaca , Vicuna , Baize , Koala , and WizardLM underwent supervised fine-tuning (SFT) on general datasets of varying sizes and complexity using the LLaMA-7B base model. MedAlpaca performed SFT on medical datasets, while Alpaca-Lora employed the same instruction dataset as Alpaca but used the Lora training method. Beaver, based on the Alpaca model, utilized RLHF alignment technology. MiniGPT-4 is a multimodal model that underwent SFT on 5 million aligned image-text pairs. Chinese-Alpaca and Chinese-LLaMA

| Model | BoolQ | HellaSwag | PIQA | WinoGrande | ARC-e | ARC-c | RACE | MMLU | Avg. |
|---|---|---|---|---|---|---|---|---|---|
| LLaMA | 75.11 | 76.19 | 79.16 | 70.00 | 72.90 | 44.80 | 40.00 | 32.75 | 61.36 |
| Alpaca | 77.49 | 75.64 | 77.86 | 67.80 | 70.66 | 46.58 | 43.16 | 41.13 | 62.54 |
| $+L_A$(epoch1) | 45.44 | 31.16 | 67.63 | 48.70 | 49.03 | 34.13 | 22.78 | 23.13 | 40.25 |
| $+L_A$(epoch2) | 42.23 | 26.09 | 49.78 | 47.43 | 26.43 | 28.92 | 22.97 | 23.22 | 33.38 |
| $+L_A$(epoch3) | 39.05 | 26.40 | 49.95 | 48.30 | 26.52 | 28.75 | 22.97 | 23.98 | 33.24 |
| $+L_A$(epoch4) | 41.62 | 26.15 | 50.11 | 49.33 | 26.56 | 28.50 | 22.78 | 23.12 | 33.52 |
| $+L_A$(epoch5) | 38.56 | 26.13 | 50.11 | 50.20 | 26.22 | 29.10 | 22.39 | 27.02 | 33.72 |

Table 4: Zero-shot performance on multiple standard benchmarks.

expanded the Chinese vocabulary based on the original LLaMA model, continuing pretraining with 20GB of text data, with Chinese Alpaca further incorporating 2M to 4.3M additional samples during SFT. Additionally, Baichuan , OpenLLaMA, InternLM , and LLaMA2 have identical architectures to LLaMA-7B but were trained from scratch by different organizations or using different training methods. We can observe that despite various and extensive subsequent training processes, as long as the LLM inherits the LLaMA-7B base model, it still exhibits very high cosine similarity w.r.t. its base model. On the other hand, the LLMs trained from scratch independently exhibit low cosine similarities w.r.t. the LLaMA-7B base model.

# D  STYLEGAN2 GENERATOR

StyleGAN2 is an improved model based on the style-based GAN architecture. One of its key enhancements is the incorporation of the perceptual path length (PPL) metric, which was originally introduced to quantify the smoothness of the mapping from the latent space to the output image. The PPL metric measures the average LPIPS distances(Zhang et al., 2018) between generated images under small perturbations in the latent space. Through the utilization of path length regularization, StyleGAN2 achieves enhanced reliability, consistency, and robustness, resulting in a smoother behavior of the generator. This regularization technique aligns with our objective of obtaining a smooth generator.

# E  DETAIL PERFORMANCE

Please refer to Table 4 for a detailed breakdown of performances on each task.

# F  CORRESPONDING SFT MODELS

Falcon-40B-Instruct: Fine-tuned based on Falcon-40B with data from Baize.

Falcon-40B-SFT-Top1-560: Fine-tuned using top-1 quality demonstrations from the OASST dataset(Köpf et al., 2023).

MPT-7B-Instruct: Fine-tuned on data derived from Databricks Dolly-15k(Conover et al., 2023) and the Anthropic Helpful and Harmless (HH-RLHF) datasets(Bai et al., 2022).

MPT-7B-StoryWriter: Optimized for reading and writing fictional stories, fine-tuned with a context length of 65k tokens.

LLaMA2-7B-Chat: Tailored for dialogue applications.

LLaMA-2-7B-32K: Trained through a two-phase process involving continued pretraining on diverse data mixtures and fine-tuning to enhance its long-context and few-shot capacity.

Qwen-7B-Chat: A large AI assistant model trained with alignment techniques.

Firefly-Qwen-7B: Fine-tuned version of Qwen-7B by the Firefly project(Yang, 2023).

Baichuan-13B-Chat: Fine-tuned and alignment-enhanced Chat version of Baichuan-13B-Base, using various instruction-following datasets.

Baichuan-13B-SFT: A bilingual instruction-tuned LoRA model of Baichuan-13B-Base, fine-tuned with instruction-following datasets.

InternLM-7B-Chat: Optimized for dialogue use cases.

Firefly-InternLM: Fine-tuned version of InternLM-7B-Chat by the Firefly project.

## G  GPT-NEOX MODELS WITH DIFFERENT GLOBAL SEEDS

To further verify the substantial variability in the distribution of model parameters during training from scratch, we conducted an experiment involving four GPT-NeoX-350M(Black et al., 2022) models trained on a subset of the Pile dataset (Gao et al., 2020). These models were trained using different global number seeds while sharing the same architecture, dataset, computational resources, and hyperparameters.

Subsequently, we computed the cosine similarities between these GPT-NeoX models, as shown in Table 2. Additionally, we employed the HuRef model to generate fingerprints for these models, depicted in Figure 8. The results revealed a noteworthy pattern: when GPT-NeoX models are trained from scratch, even minor changes to the global seed lead to cosine similarities approaching zero. Correspondingly, their fingerprints exhibited clear distinctions from each other.

| ICS | Seed=1 | Seed=2 | Seed=3 | Seed=4 |
|---|---|---|---|---|
| Seed=1 | 100.00 | 14.12 | 6.13 | 5.72 |
| Seed=2 | 14.12 | 100.00 | 6.52 | 5.97 |
| Seed=3 | 6.13 | 6.52 | 100.00 | 12.71 |
| Seed=4 | 5.72 | 5.97 | 12.71 | 100.00 |

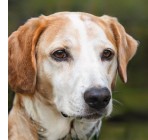 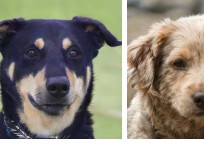 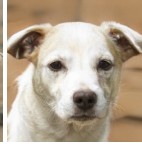

Seed1     Seed2     Seed3     Seed4

Table 5: ICS values between GPT-NeoX models with different global seeds

Figure 8: Fingerprints of GPT-NeoX models trained with varying global seeds

## H  OPEN-SOURCED LLMS

In these experiments, we aimed to gather diverse models covering various parameter sizes. For the widely used LLaMA models, we included LLaMA-7B, LLaMA-13B, LLaMA-65B, and LLaMA2-7B. We also incorporated models with similar architectures to LLaMA, such as InternLM-7B, OpenLLaMA-7B, and Baichuan-7B. To encompass a broader range of parameters, we expanded our collection to include GPT2-Large(Radford et al., 2019), Cerebras-GPT-1.3B(Dey et al., 2023), GPT-Neo-2.7B, and even the largest Falcon-180B. Additionally, we considered models like Qwen-7B, RedPajama-7B(Computer, 2023), ChatGLM-6B(Du et al., 2022), Bloom-7.1B(Workshop et al., 2022), ChatGLM2-6B,Pythia-6.9B and 12B(Biderman et al., 2023), OPT-6.7B and 30B, and GPT-NeoX-20B(Black et al., 2022), among other commonly used LLMs. In total, we gathered a representative set of 24 open-sourced LLMs.

## I  ICS OF LLMS WITH DIFFERENT PARAMETER SIZES

| ICS | LLaMA | Qwen | LLaMA2 | RedP | CGLM | Bloom | CGLM2 | Pythia | InternLM | OpenLA | OPT | Baichuan |
|---|---|---|---|---|---|---|---|---|---|---|---|---|
| LLaMA | 100.00 | 1.72 | 10.71 | -0.08 | -0.12 | -0.31 | 0.10 | -0.13 | 1.05 | 2.11 | -20.90 | 2.76 |
| Qwen | 1.72 | 100.00 | 1.44 | -0.02 | 0.00 | -0.18 | 0.11 | 0.00 | 0.81 | 0.59 | -2.35 | 0.80 |
| LLaMA2 | 10.71 | 1.44 | 100.00 | -0.33 | 0.05 | -0.31 | 0.03 | -0.20 | 0.46 | 2.39 | -21.64 | 2.79 |
| RedP | -0.08 | -0.02 | -0.33 | 100.00 | -19.11 | 6.91 | 4.20 | 6.18 | -0.10 | -0.02 | 7.98 | -0.06 |
| CGLM | -0.12 | 0.00 | 0.05 | -19.11 | 100.00 | 2.16 | -5.30 | -3.34 | -0.02 | 0.02 | -15.82 | 0.02 |
| Bloom | -0.31 | -0.18 | -0.31 | 6.91 | 2.16 | 100.00 | 2.34 | 2.15 | -0.08 | -0.01 | 2.70 | -0.12 |
| CGLM2 | 0.10 | 0.11 | 0.03 | 4.20 | -5.30 | 2.34 | 100.00 | 1.86 | 0.44 | 0.10 | 1.20 | 0.06 |
| Pythia | -0.13 | 0.00 | -0.20 | 6.18 | -3.34 | 2.15 | 1.86 | 100.00 | -0.02 | -0.04 | 1.97 | -0.07 |
| InternLM | 1.05 | 0.81 | 0.46 | -0.10 | -0.02 | -0.08 | 0.44 | -0.02 | 100.00 | 0.77 | -7.24 | 0.49 |
| OpenLA | 2.11 | 0.59 | 2.39 | -0.02 | 0.02 | -0.01 | 0.10 | -0.04 | 0.77 | 100.00 | -5.72 | 0.88 |
| OPT | -20.90 | -2.35 | -21.64 | 7.98 | -15.82 | 2.70 | 1.20 | 1.97 | -7.24 | -5.72 | 100.00 | -7.22 |
| Baichuan | 2.76 | 0.80 | 2.79 | -0.06 | 0.02 | -0.12 | 0.06 | -0.07 | 0.49 | 0.88 | -7.22 | 100.00 |

Table 6: The cosine similarities of invariant terms(ICS) between different base LLMs. Abbreviations used include "RedP" for "RedPajama," "OpenLA" for "OpenLLaMA," "CGLM" and "CGLM2" representing "ChatGLM" and "ChatGLM2," respectively. Additionally, except for the ChatGLM and ChatGLM2, all other models have a parameter size of approximately 7B.

| ICS | Pythia-12B | LLaMA-13B |
|---|---|---|
| Pythia-12B | 100.00 | 0.03 |
| LLaMA-13B | 0.03 | 100.00 |

Table 7: ICS between similar size LLMs:Pythia-12B and LLaMA-13B.

| ICS | OPT-30B | LLaMA-30B | Galactica-30B | GPT-NeoX-20B |
|---|---|---|---|---|
| OPT-30B | 100.00 | -4.23 | -32.70 | 2.76 |
| LLaMA-30B | -4.23 | 100.00 | -0.40 | -0.02 |
| Galactica-30B | -32.70 | -0.40 | 100.00 | -4.27 |
| GPT-NeoX-20B | 2.76 | -0.02 | -4.27 | 100.00 |

Table 8: ICS between LLMs with their sizes ranging from 20B to 30B:GPT-NeoX-20B, OPT-30B, LLaMA-30B and Galactica-30B.

| ICS | Falcon-180B | LLaMA-65B | Galactica-120B |
|---|---|---|---|
| Falcon-180B | 100.00 | 0.02 | 20.99 |
| LLaMA-65B | 0.02 | 100.00 | -0.10 |
| Galactica-120B | 20.99 | -0.10 | 100.00 |

Table 9: ICS between LLMs with their sizes not less than 65B: LLaMA-65B, Galactica-120B and Falcon-180B.

| ICS | GPT2 | CGPT1.3 | Neo2.7 | CGLM | CGLM2 | OPT6.7 | Pythia6.9 | LM7 | Qwen | LM27 | RedPaj | Bloom | Internlm | OLM | Baichuan | Pythia12 | LM13 | Neox20 | OPT30 | LM30 | Gala30 | LM65 | Gala120 | Fal180 |
|---|---|---|---|---|---|---|---|---|---|---|---|---|---|---|---|---|---|---|---|---|---|---|---|---|
| GPT2 | 100.00 | 32.59 | -14.72 | 79.10 | -4.01 | 27.34 | -1.79 | -9.34 | -0.93 | -9.90 | -12.16 | 5.85 | -3.13 | -2.49 | -3.05 | -1.51 | -3.53 | -4.10 | -10.13 | -4.83 | 66.45 | -3.26 | -71.75 | -9.96 |
| CGPT1.3 | 32.59 | 100.00 | 77.73 | 4.13 | 3.97 | 43.78 | 5.00 | -7.34 | -0.79 | -8.49 | 24.50 | 18.50 | -2.51 | -2.01 | -2.55 | 4.63 | -2.72 | 1.07 | 46.75 | -3.46 | 2.20 | -2.42 | -35.80 | -47.46 |
| Neo2.7 | -14.72 | 77.73 | 100.00 | -31.70 | 6.04 | 12.73 | 6.01 | 3.10 | 0.70 | 2.44 | 31.83 | 17.55 | -1.35 | 0.46 | 0.95 | 5.72 | 0.92 | 3.07 | 41.93 | 1.14 | -28.56 | 2.11 | -6.54 | -47.12 |
| Chatglm | 79.10 | 4.13 | -31.70 | 100.00 | -5.30 | -15.82 | -3.34 | -0.12 | 0.00 | 0.05 | -19.11 | 2.16 | -0.02 | 0.02 | 0.02 | -2.66 | 0.18 | -5.10 | -38.77 | -0.02 | 77.23 | -0.05 | -77.36 | -1.38 |
| Chatglm2 | -4.01 | 3.97 | 6.04 | -5.30 | 100.00 | 1.20 | 1.86 | 0.10 | 0.11 | 0.03 | 4.20 | 2.34 | 0.44 | 0.10 | 0.06 | 1.80 | 0.10 | 1.30 | 3.79 | -0.17 | -4.62 | -0.15 | 2.64 | -2.75 |
| OPT6.7 | 27.34 | 43.78 | 12.73 | -15.82 | 1.20 | 100.00 | 1.97 | -20.90 | 0.00 | -21.64 | 7.98 | 2.70 | -7.24 | -5.72 | -7.22 | 1.08 | -7.25 | 1.16 | 54.00 | -8.96 | -12.71 | -7.06 | 9.21 | -7.63 |
| Pythia6.9 | -1.79 | 5.00 | 6.01 | -3.34 | 1.86 | 1.97 | 100.00 | -0.13 | 0.00 | 0.00 | 6.18 | 2.15 | -0.02 | -0.04 | -0.07 | 3.32 | -0.04 | 1.88 | 3.79 | -0.12 | -2.99 | -0.08 | 0.77 | -3.15 |
| LM7 | -9.34 | -7.34 | 3.10 | -0.12 | 0.10 | -20.90 | -0.13 | 100.00 | 1.72 | 10.71 | -0.08 | -0.31 | 1.05 | 2.11 | 2.76 | 0.06 | 2.80 | -0.02 | -7.46 | 2.72 | -0.40 | 2.91 | -0.22 | 0.11 |
| Qwen | -0.93 | -0.79 | 0.70 | 0.00 | 0.11 | 0.00 | 0.00 | 1.72 | 100.00 | 1.44 | -0.02 | -0.18 | 0.81 | 0.59 | 0.80 | 0.03 | 1.03 | -0.01 | -0.77 | -0.64 | -0.01 | 0.31 | -0.06 | 0.01 |
| LM27 | -9.90 | -8.49 | 2.44 | 0.05 | 0.03 | -21.64 | 0.00 | 10.71 | 1.44 | 100.00 | -0.33 | -0.31 | 0.46 | 2.39 | 2.79 | 5.60 | 2.68 | 3.65 | -8.48 | 4.46 | -16.58 | 3.57 | 5.51 | 0.63 |
| RedPaj | -12.16 | 24.50 | 31.83 | -19.11 | 4.20 | 7.98 | 6.18 | -0.08 | -0.02 | -0.33 | 100.00 | 6.91 | -0.10 | -0.02 | -0.06 | 5.60 | -0.06 | 3.65 | 18.21 | 0.09 | -16.58 | 0.01 | 5.51 | -15.12 |
| Bloom | 5.85 | 18.50 | 17.55 | 2.16 | 2.34 | 2.70 | 2.15 | -0.31 | -0.18 | -0.31 | 6.91 | 100.00 | -0.08 | -0.01 | -0.12 | 2.12 | -0.24 | 0.98 | 6.39 | 0.61 | 1.86 | 0.02 | -9.84 | -11.48 |
| Internlm | -3.13 | -2.51 | -1.35 | -0.02 | 0.44 | -7.24 | -0.02 | 1.05 | 0.81 | 0.46 | -0.10 | -0.08 | 100.00 | 0.77 | 0.49 | 0.04 | 0.78 | 0.00 | -2.46 | -1.46 | 0.09 | -1.07 | -0.05 | 0.02 |
| OLM | -2.49 | -2.01 | 0.46 | 0.02 | 0.10 | -5.72 | -0.04 | 2.11 | 0.59 | 2.39 | -0.02 | -0.01 | 0.77 | 100.00 | 0.88 | 0.02 | 0.88 | -0.00 | -2.07 | -0.05 | -0.01 | 0.71 | -0.09 | 0.02 |
| Baichuan | -3.05 | -2.55 | 0.95 | 0.02 | 0.06 | -7.22 | -0.07 | 2.76 | 0.80 | 2.79 | -0.06 | -0.12 | 0.49 | 0.88 | 100.00 | 0.02 | 1.06 | -0.02 | -2.45 | 0.96 | -0.07 | 1.01 | -0.11 | -0.07 |
| Pythia12 | -1.51 | 4.63 | 5.72 | -2.66 | 1.80 | 1.08 | 3.32 | 0.06 | 0.03 | 5.60 | 5.60 | 2.12 | 0.04 | 0.02 | 0.02 | 100.00 | 0.03 | 1.82 | 3.03 | -0.00 | -2.50 | 0.01 | 0.31 | -3.10 |
| LM13 | -3.53 | -2.72 | 0.92 | 0.18 | 0.10 | -7.25 | -0.04 | 2.80 | 1.03 | 2.68 | -0.06 | -0.24 | 0.78 | 0.88 | 1.06 | 0.03 | 100.00 | -0.02 | -3.22 | 0.68 | 0.07 | 1.51 | -0.25 | 0.07 |
| Neox20 | -4.10 | 1.07 | 3.07 | -5.10 | 1.30 | 1.16 | 1.88 | -0.02 | -0.01 | 3.65 | 3.65 | 0.98 | 0.00 | -0.00 | -0.02 | 1.82 | -0.02 | 100.00 | 2.76 | -0.02 | -4.27 | -0.02 | 3.54 | -0.84 |
| OPT30 | -10.13 | 46.75 | 41.93 | -38.77 | 3.79 | 54.00 | 3.79 | -7.46 | -0.77 | -8.48 | 18.21 | 6.39 | -2.46 | -2.07 | -2.45 | 3.03 | -3.22 | 2.76 | 100.00 | -4.23 | -32.70 | -2.82 | 20.12 | -18.06 |
| LM30 | -4.83 | -3.46 | 1.14 | -0.02 | -0.17 | -8.96 | -0.12 | 2.72 | -0.64 | 4.46 | 0.09 | 0.61 | -1.46 | -0.05 | 0.96 | -0.00 | 0.68 | -0.02 | -4.23 | 100.00 | -0.40 | 4.53 | -0.07 | 0.06 |
| Gala30 | 66.45 | 2.20 | -28.56 | 77.23 | -4.62 | -12.71 | -2.99 | -0.40 | -0.01 | -0.32 | -16.58 | 1.86 | 0.09 | -0.01 | -0.07 | -2.50 | 0.07 | -4.27 | -32.70 | -0.40 | 100.00 | -0.27 | -79.95 | -0.23 |
| LM65 | -3.26 | -2.42 | 2.11 | -0.05 | -0.15 | -7.06 | -0.08 | 2.91 | 0.31 | 3.57 | 0.01 | 0.02 | -1.07 | 0.71 | 1.01 | 0.01 | 1.51 | -0.02 | -2.82 | 4.53 | -0.27 | 100.00 | -0.10 | 0.02 |
| Gala120 | -71.75 | -35.80 | -6.54 | -77.36 | 2.64 | 9.21 | 0.77 | -0.22 | -0.06 | -0.01 | 5.51 | -9.84 | -0.05 | -0.09 | -0.11 | 0.31 | -0.25 | 3.54 | 20.12 | -0.07 | -79.95 | -0.10 | 100.00 | 20.99 |
| Fal180 | -9.96 | -47.46 | -47.12 | -1.38 | -2.75 | -7.63 | -3.15 | 0.11 | 0.01 | 0.63 | -15.12 | -11.48 | 0.02 | 0.02 | -0.07 | -3.10 | 0.07 | -0.84 | -18.06 | 0.06 | -0.23 | 0.02 | 20.99 | 100.00 |

Table 10: ICS between 24 open-sourced LLMs with sizes ranging from 774M to 180B.Abbreviations used include "GPT2" for "GPT2-Large," "CGPT1.3" for "Cerebras-GPT-1.3B," "Neo2.7" for "GPT-Neo-2.7B," "CGLM" for "ChatGLM-6B," "CGLM2" for "ChatGLM-6B2," "OPT6.7" for "OPT6.7B," "Pythia6.9" for "Pythia-6.9B," "LM7" for "LLaMA-7B," "Qwen" for "Qwen-7B," "LM27" for "LLaMA2-7B," "RedPaj" for "RedPajama-7B," "Bloom" for "Bloom-7B," "Internlm" for "InternLM-7B," "OLM" for "OpenLLaMA-7B," "Baichuan" for "Baichuan-7B," "Pythia12" for "Pythia-12B," "LM13" for "LLaMA-13B," "Neox20" for "GPT-NeoX-20B," "OPT30" for "OPT-30B," "LM30" for "LLaMA-30B," "Gala30" for "Galactica-30B," "LM65" for "LLaMA-65B," "Gala120" for "Galactica-120B," and "Fal180" for "Falcon-180B" respectively.From left to right LLMs are sorted by parameter size from smallest to largest.

