# OpenReview forum: "HuRef: HUman-REadable Fingerprint  for Large Language Models"
_ICLR.cc/2024/Conference — Submitted to ICLR 2024_

### Official Review · Reviewer_feP3 · 2023-10-25

**Soundness:** 2 fair
**Presentation:** 2 fair
**Contribution:** 2 fair
**Rating:** 3
**Confidence:** 4

**Summary:**

The paper starts with an observation that the vector direction of LLM parameters can be a good fingerprint to identify the original base model. However, doing this will expose the model parameters and is not robust to attacks like linear mapping or permutation on model parameters or word embeddings. Instead, the paper proposes the invariant terms that are sharable by the model owners, more robust to the above attacks, but sacrificing a small amount of performance. It then converts the invariant terms to human-readable fingerprints using a conditional StyleGAN2 to generate dog images. Experimental results demonstrated the effectiveness of the proposed method when the LLMs undergo different training steps like SFT, RLHF, or fine-tuning.

**Strengths:**

+ The proposed invariant terms are novel and effective in identifying the variances of the LLMs.
+ The idea of using visual information for explanation is creative.

**Weaknesses:**

+ The paper has a citation format issue (possibly by converting from an IEEE-formatted source).
+ The paper lacks a solid theoretical foundation for developing the proposed method.
+ The proposed invariant terms may not work with the distillation attacks where the original LLMs are used as a teacher for training. The student may have a different architecture than the teacher.
+ Mapping from invariant terms to dog images relies too much on the disentanglement quality of StyleGAN2. For example, in Fig. 5, the Guanaco dog differs from the others. In Fig. 6, there are some similar but unrelated pairs, such as [Qwen-7B, Galactica-30B], [ChatGLM-6B, LLaMA-13B], [Bloom-7B, OPT-30B], and [Baichuan-7B, Pythia-12B].
+ The current training scheme for the convolutional encoder may lead to overfitting. There is no evidence of the separation of training and test data, which should be mutually exclusive.

**Questions:**

Please refer to the comments in the weaknesses section.

---

> ### Author Response · Authors · 2023-11-19
>
> Thank you for acknowledging the innovation and effectiveness of our research and providing numerous constructive suggestions. We sincerely appreciate your time spent reviewing the paper, and we provide detailed responses to each of your comments below.
>
>
> > 1. The paper has a citation format issue (possibly by converting from an IEEE-formatted source).
>
> Thanks for pointing this out. We have updated the citation format in the newly updated paper.
>
> > 2. The paper lacks a solid theoretical foundation for developing the proposed method.
>
> Our proposed method is based on the observation that the vector direction of parameters of LLMs, after undergoing SFT, RLHF, and even continued pretraining, do not perturb significantly. The close relation between the parameter vector direction and the base model is validated both in the sufficiency and necessity aspects. For sufficiency, we validate that comparing the vector direction between two models is sufficient to detect if they share the same base model. More importantly, for necessity, we show that a potential attacker can't deviate from the vector direction without damaging the base model's abilities (Figure 1).
>
> We want to mention the classical RSA algorithm in cryptography as an analogy. Although there is no theoretical guarantee that there do not exist fast algorithms for large integer factorization, all we know is that practically we do not have known efficient algorithms given current computation devices. The lack of theoretical guarantee doesn't stop RSA from becoming the classical algorithm in cryptography. Our work shows that (1) the parameter vector direction is closely related to the base model and is pretty stable under subsequent training steps, and (2) there is no cheap way for attackers to circumvent this vector direction test while still preserving the base model's pretrained abilities.
>
> > 3. The proposed invariant terms may not work with distillation attacks where the original LLMs are used as a teacher for training. The student may have a different architecture than the teacher.
>
> Our work focuses on identifying the base model to prevent theft and misuse on the base model. Training a student model with a different architecture, probably from scratch, by distilling the original LLM requires a substantial amount of data and computation that prohibits people from doing it. As for distilling SFT data from LLMs, it still requires a base model to apply SFT. In this work we focus on protecting the base model from misuse, so protecting LLMs from SFT data distillation is out of our scope.
>
> > 4. Mapping from invariant terms to dog images relies too much on the disentanglement quality of StyleGAN2. For example, in Fig. 5, the Guanaco dog differs from the others. In Fig. 6, there are some similar but unrelated pairs, such as [Qwen-7B, Galactica-30B], [ChatGLM-6B, LLaMA-13B], [Bloom-7B, OPT-30B], and [Baichuan-7B, Pythia-12B].
>
> Thanks for your careful observation. We do concede that StyleGAN2 is not a perfect visualization generation model for our invariant terms under our situation, which causes several unwanted similar images. We would like to emphasize that for a quantitative and objective comparison, the invariant terms that we derived are the things to rely on. Our motivation to put an image generator is only to provide a vivid visualization of the terms, which could intuitively reflect the similarity between the models being compared. As for the generated images, we can tell that StyleGAN2-generated images are far more similar between models sharing the same base model than those that are not. We are actually working on a specialized image generation model as our subsequent research. Given that the current StyleGAN2 is fairly good, and the 9-page limit is already very tight to describe our current contents, we believe the design of the specialized generator is beyond the scope of this research.
>
> As for the similar images between [Baichuan-7B, Pythia-12B], we find that it is our bad. We mistakenly put the Baichuan-7B image for Pythia-12B while we were generating the figure. We have updated the figure in the new version of the paper.
>
>
> > 5. The current training scheme for the convolutional encoder may lead to overfitting. There is no evidence of the separation of training and test data, which should be mutually exclusive.
>
> We want to clarify that the convolutional encoder is merely trained by synthetic random matrices only, without being exposed to any real LLM-derived invariant terms (c.f. Section 4.1 and Figure 3). In this case, we strictly ensure mutual exclusivity between training and test data. We have also added a new sentence (highlighted in Section 4.1) and a new subsection detailed data synthesis (c.f. Appendix A.1) to emphasize this point.

---

> > ### Comment · Reviewer_feP3 · 2023-11-21
> >
> > I would like to thank the authors for their responsiveness in addressing my inquiries and for revising the paper. While I appreciate the effort put into the manuscript, regrettably, I must maintain my reservations about the StyleGAN2 section. I wish the authors every success in their subsequent submission with their further improvements.

---

> > > ### Author Response · Authors · 2023-11-22
> > >
> > > Thank you for your response. We will continue to improve our work. Wishing you all the best in your endeavors.

---

> ### Author Response · Authors · 2023-11-21
> **The end of the discussion phase approaching**
>
> Dear Reviewer feP3, as the discussion period ends soon, we would like to check whether our responses answer your questions. Following your comments, we have updated the citation format and emphasized the mutual separation of training and test data in the newly updated paper. The other questions received a direct response as well. Thank you again for your comments and suggestions to improve our paper, and we look forward to your reply.

---

### Official Review · Reviewer_qUfX · 2023-11-01

**Soundness:** 3 good
**Presentation:** 3 good
**Contribution:** 2 fair
**Rating:** 5
**Confidence:** 3

**Summary:**

This paper aims at identifying the original base model of an LLMs which are fine-tuned or even continue-pretrained.
The paper first points out that for the LLMs from the same baseline model, the cosine similarity would be close.
Based on this observation, the invariant terms are proposed to represent a LLM considering that the actual parameters are accessible in some scenarios.

**Strengths:**

- The paper targets at an interesting topic which has not been explored thoroughly.
- Using the invarient term to represent a network sounds reasonable.

**Weaknesses:**

- For the experiment showing that LLM models from a same base model would have a higher cosine similarity, this result is quite predictable as some models are finetuned on only several modules instead of the whole network. Have you also tried to compute another similarity score? It would be interesting whether a simple Euclidean distance could lead to the same result or not.
- Representing a LLM into a readable dog image is an interesting idea, but is not practical and scientific. The visual similarity is subjective. For example, Guanaco and LLaMA in Figure 5 looks more different than Qwen-7B and Galactica-30B in Figure 6.
- Given two concerns mentioned above, the contributions of this paper are limited and I doubt this work can bring the impact the community.

**Questions:**

- What are the points to use a figure instead of numerical difference to represent a LLM? Could you please specify some advantages?
- Have you tried to use different similarity functions for the experiment shown in Table 1?

---

> ### Author Response · Authors · 2023-11-19
>
> Thank you for your acknowledgment of our research topic and method's reasonableness, as well as for providing helpful suggestions. We sincerely appreciate your time in reading the paper, and our point-to-point responses to your comments are given below.
>
> > 1. For the experiment showing that LLM models from the same base model would have a higher cosine similarity, this result is quite predictable, as some models are fine-tuned on only several modules instead of the whole network.
>
> These experiments included not only fine-tuning with LoRA but also using SFT, RLHF, or multimodal alignment(MiniGPT-4 underwent SFT on 5 million aligned image-text pairs) for updating the entire set of parameters. We even explored expanding the token list and continuing pretraining on large Chinese datasets(Chinese-Alpaca and Chinese-LLaMA continue pretrained with 20GB of text data). These experiments illustrate that our observation holds when updating the entire parameter set, and is stable after the model has undergone a large amount of training data in subsequent training steps.
>
> > 2. Have you also tried to compute another similarity score? It would be interesting whether a simple Euclidean distance could lead to the same result or not.
>
> This is an excellent question. Following your suggestion, we added experimental results using Euclidean distance (ED) and Minkowski Distance (MD,$p=3$) to calculate relative distances.
> $$MD(\mathbf{X}, \mathbf{Y}) = \left( \sum\_{i=1}^{n} |X\_i - Y\_i|^3 \right)^{\frac{1}{3}}$$
> P-ED and I-ED represent Euclidean distances calculated from model parameters and model invariant terms, respectively. P-MD and I-MD represent Minkowski Distance ($p=3$) calculated from model parameters and model invariant terms, respectively:
>
> | Model | Alpaca-Lora | Alpaca | Chinese-LLaMA | Vicuna | Baize | MedAlpaca | Koala | WizardLM | MiniGPT-4 | Chinese-Alpaca |*Baichuan* | *OpenLLaMA* | *InternLM* | *LLaMA-2* |
> |-------|-------------|--------|---------------|--------|-------|-----------|-------|----------|-----------|----------------|----------|-----------|----------|---------|
> | P-ED   |      467       | 577|1246|1364|1567|730|1297|782|1866|2286|*42298*|*38290*|*44760*|*35492*|
> | I-ED   | 60.8 | 16.1 | 237.7 | 58.0 | 52.5 | 27.7 | 44.7 | 24.5 | 66.1 | 270.9 | *1938.1* | *903.8* | *1373.1* | *639.8*|
> | P-MD   |  38.2|39.2|103.7|92.5|110.9|49.6|100.0|53.1|127.3|176.0|*2914.3*|*2639.9*|*3122.1*|*2463.7*|
> | I-MD   | 4.3 | 1.0 | 17.7 | 3.5 | 3.5 | 1.7 | 2.7 | 1.5 | 3.9 | 19.4 | *129.1* | *54.2* | *84.8* | *38.7* |
>
> These results align with previous experiments, still showing substantial differences in distances between the same models and models from different sources. We chose cosine similarity in our paper because it is well-normalized w.r.t. number of dimensions in the vector, vector norms, etc., have a clear geometric meaning
> , well-bounded between [-1, 1], and is widely used in the NLP field.
>
> > 3. Representing an LLM as a readable dog image is an interesting idea, but is not practical and scientific. The visual similarity is subjective. For example, Guanaco and LLaMA in Figure 5 look more different than Qwen-7B and Galactica-30B in Figure 6.
>
>
> We aim to utilize a readable dog image to provide a vivid visualization for invariant terms, allowing for an intuitive, qualitative reflection of the similarity between the compared models. In most cases, we can observe that StyleGAN2-generated images are significantly more similar among models sharing the same base model than those that do not. In case it becomes vague to subjectively determine the similarity between images, we can always roll back and rely on the cosine similarity between invariant terms. We also concede that although StyleGAN2 generator performs fairly well in our experiments it is still not an ideal generator for some rare cases, which suggests that a specially designed image generator could yield better images for qualitative comparison. We are indeed doing this, but considering the page limits of this single paper, we plan to further address this issue in our upcoming work.
>
>
> > 4. What are the points to use a figure instead of numerical difference to represent an LLM? Could you please specify some advantages?
>
> Compared to invariant terms, using a dog figure is a more vivid representation of the model weights, and thus it is more human-readable. It could provide a nice visualization of the model weights that allows almost all people to recognize while comparing the invariant terms requires people to be familiar with related knowledge in the domain of deep learning. We can directly discover the base model of an LLM merely by looking at the images. We believe that images and invariant terms can be complementary rather than mutually exclusive, as each has its own advantages.
>
> > 5. Have you tried using different similarity functions for the experiment shown in Table 1?
>
> Please refer to our response to comment 2.

---

> > ### Comment · Reviewer_qUfX · 2023-11-23
> >
> > I thank the authors for the response. I have checked the author rebuttal and other reviewer's feedback. The concern about not too much performance improvement compared to StyleGAN2 generator is not fully addressed. Besides, I suggest using numerical difference as complementary to the figure to represent an LLM. Thus, I maintain my current rating for the paper and hope the authors can fully address the issue in the upcoming work.

---

> ### Author Response · Authors · 2023-11-21
> **The end of the discussion phase approaching**
>
> Dear Reviewer qUfX, as the discussion period draws to a close, we would like to verify if our responses adequately address your inquiries. In light of what you've mentioned, we conducted experiments using different similarity functions.  The other questions received a direct response as well. We greatly appreciate your valuable feedback and recommendations to enhance our paper. We eagerly await your response.

---

### Official Review · Reviewer_8z4E · 2023-11-01

**Soundness:** 4 excellent
**Presentation:** 4 excellent
**Contribution:** 4 excellent
**Rating:** 6
**Confidence:** 3

**Summary:**

This paper introduces a novel approach called HuRef, which is a human-readable fingerprint for large language models (LLMs) that uniquely identifies the base model without exposing model parameters or interfering with training. The authors observe that the vector direction of LLM parameters remains stable after the model has converged during pretraining, and this stability can be used to identify the base model. They also address vulnerability to attacks by defining three invariant terms that identify an LLM's base model. The authors then propose a fingerprinting model that maps these invariant terms to a Gaussian vector and converts it into a natural image using an off-the-shelf image generator. Experimental results demonstrate the effectiveness of the proposed method.

**Strengths:**

1) The paper presents an innovative and practical methodology for identifying the base model of LLMs without exposing model parameters.
2) The authors provide insightful empirical findings by showing the stability of the vector direction of LLM parameters and the effectiveness of the proposed invariant terms.
3) The paper is well-structured and provides a clear review of relevant literature.

**Weaknesses:**

1) The paper lacks detailed implementation details, making it difficult to reproduce the study.
2) The authors should provide more in-depth insights into why the proposed method is effective.

**Questions:**

1) Could you provide more details about the implementation of the proposed method, including the specific architecture and training settings?
2) Can you clarify how the proposed method can be applied to LLMs that are not open-sourced or have restricted access to their parameters?

---

> ### Author Response · Authors · 2023-11-19
>
> Thank you for acknowledging that our method is innovative and practical, and for providing constructive comments. We sincerely appreciate your time in reading the paper, and our point-to-point responses to your comments are given below.
>
> > 1. The paper lacks detailed implementation details, making it difficult to reproduce the study.
>
> We are sorry for the confusion. Actually, we do want to expose as many detailed implementations as possible. To ensure reproducibility, we have committed code in the first submission and we plan to open-source our code in the camera-ready version. During the rebuttal period, we have added a comprehensive explanation of the implementation details in the appendix (c.f. Appendix A).
>
> > 2. The authors should provide more in-depth insights into why the proposed method is effective.
>
> Our experiments in Sections 3.1.1 and 3.1.2 provide evidence of the reliability and effectiveness of our method in two folds. First, the model parameters' vector direction is closely related to the base model, subsequent training steps (such as SFT, RLHF, or continued pretraining) won't change it significantly. Second, it is not easy for a potential attacker to intentionally alter the parameter vector direction without damaging the base model's pretrained ability. This underlies the foundation of the reliability of our proposed method.
>
> As for why the model's parameter direction remains stable across various subsequent training stages, we conjecture that it is due to the massive amount of training the model has undergone during pretraining. The unique parameter vector direction of a trained model can be ultimately traced back to its random initialization. As long as the models are initialized independently, their vector directions could be completely different despite that the training procedures and data are identical (c.f. our experiments in Appendix G). During pretraining, as the model converges, the vector direction of the model also stabilizes gradually. Once it stabilizes, the vector direction won't change too much unless it is intentionally altered, which results in major damage to the model (c.f. Figure 1, and Section 3.1.2).
>
> In addition, we have also added a new experiment to show how the model's parameter vector direction stabilizes during pretraining, by comparing neighboring checkpoints of a model during its pretraining and calculating the cosine similarities. As pretraining progresses, we observe a diminishing change in the model's parameter direction. Due to time and computational constraints, we trained a GPT-Neox model with 350 million parameters for 360,000 steps and saved a checkpoint every 50K steps. For larger models and more pretraining steps, we expect this phenomenon to be more pronounced.
>
> | Comparing CKPTs  | 10K-60K | 60k-110k | 110k-160k | 160k-210k | 210k-260k | 260k-310k | 310k-360k |
> |--------|---------|----------|-----------|-----------|-----------|-----------|-----------|
> | Cossim | 56.23   | 84.65    | 88.95     | 90.96     | 92.13     | 93.22     | 94.25     |
>
>
> > 3. Could you provide more details about the implementation of the proposed method, including the specific architecture and training settings?
>
> Of course, we have added a section in the appendix(c.f. Appendix A) that provides a detailed explanation of the implementation of our proposed method, especially the specific architecture and training settings we employed.
>
> > 4. Can you clarify how the proposed method can be applied to LLMs that are not open-sourced or have restricted access to their parameters?
>
> Our methods do not require the LLM manufacturers to release the model parameters in order to provide proof to protect their model. In fact, only the invariant terms need to be provided. Since the invariant terms are used both for protecting their own model and for protecting models released by others, the manufacturers have to release their invariant terms responsibly. In addition, in cases of disputes or when necessary, we can also introduce third-party regulators to interfere and verify the authenticity of invariant terms, preventing manufacturers from releasing false terms. In both cases, no model parameters are needed to be released to the public or another LLM manufacturer in order to finish the verification.  We've also provided an explanatory figure and more detailed explanations in Appendix B.

---

> ### Author Response · Authors · 2023-11-21
> **The end of the discussion phase approaching**
>
> Dear Reviewer 8z4E, as the discussion period comes to a close, we would like to thank you once again for your positive assessment. Following your comments, we have added a section detailing implementations in the appendix (c.f. Appendix A). The other questions received a direct response as well. Your support and inspirational comments have been invaluable to us. We remain open and eager to incorporate any further feedback or insights you might offer.

---

> > ### Comment · Reviewer_8z4E · 2023-11-22
> >
> > Thank you for your explanation of the insights in this paper and the ablation experiments on model parameters. This has resolved my confusion, and I will maintain my rating

---

> > > ### Author Response · Authors · 2023-11-22
> > >
> > > Thank you very much for your response. I'm glad we could address your concerns. Once again, I appreciate the time you dedicated to the review. Wishing you all the best.

---

### Meta-Review · Area_Chair_xnhw · 2023-12-06

**Metareview:**

The submission "HuRef: HUman-REadable Fingerprint for Large Language Models" describes an approach toward fingerprinting of modern LLMs. The submission trains an encoder model that converts model weights into a reduced vector representation, which is argued to be invariant to a number of attacks and modifications of the weights. Finally, the authors also visualize the vector representation of each model as a generated image by using it as initial latent feature for a StyleGan2 model.

The proposed approach to model fingerprinting is certainly interesting, and the visualization an inventive cherry on top, but ultimately reviewers were not convinced by the presentation of results and experimental setup provided in this work. Several reviewers bring up concerns regarding the quality of the StyleGan visualization, with which I disagree, yet these reviews show that the core of the idea presented by the authors is not well captured in the current submission.

As such,  I do not consider this submission ready for publication just yet. I encourage the authors to revise their manuscript for submission to a future conference round, focusing more of their experimental section on comparisons to baseline fingerprinting approaches and on large-scale evaluations of the proposed fingerprinting (to be precise, what are TP/FP rates for an actual deployment of this fingerprint to a large dataset of huggingface checkpoints?). I would suggest putting more emphasis on validating the strength of the fingerprint compared to the currently large emphasis on the visualization of the fingerprint. If the authors want to keep a large focus on the visualization aspect and human readability, then this should be quantified in a human subject study.

**Justification For Why Not Higher Score:**

The presentation of results and experimental design provided in this work do not sufficiently support the core idea of this submission.

**Justification For Why Not Lower Score:**

N/A

---

### Decision · Program_Chairs · 2024-01-16

Reject